# Phytochemical Profiling and Biological Evaluation of *Dianthus sylvestris* subsp. *aristidis*: A Chromatographic and Mass Spectrometry Approach to Uncovering Bioactive Metabolites for Dermatological and Metabolic Disorder Management

**DOI:** 10.3390/ph18040578

**Published:** 2025-04-16

**Authors:** Amina Bouzana, Zohra Chekroud, Imène Becheker, Fatima Kamah, Nora Sakhraoui, Chawki Bensouici, Fehmi Boufahja, Sulaiman A. Alsalamah, Mohammed I. Alghonaim, Stefania Garzoli, Hamdi Bendif

**Affiliations:** 1Laboratory of Interactions, Biodiversity, Ecosystems and Biotechnology, Department of Nature and Life Sciences, Faculty of Sciences, University 20 August 1955 Skikda, Skikda 21000, Algeria; a.bouzana@univ-skikda.dz (A.B.); chekroudzohra@yahoo.fr (Z.C.); i.becheker@univ-skikda.dz (I.B.); kamah@univ-skikda.dz (F.K.); 2Laboratory of Interactions, Biodiversity, Ecosystems and Biotechnology, Department of Ecology and Environment, Faculty of Sciences, University 20 August 1955 Skikda, Skikda 21000, Algeria; sakhraouinora05@gmail.com; 3Biotechnology Research Center, Ali Mendjli New City UV 03, BP E73, Constantine 25016, Algeria; chawkiislam@yahoo.fr; 4Biology Department, College of Science, Imam Mohammad Ibn Saud Islamic University (IMSIU), Riyadh 11623, Saudi Arabia; faboufahja@imamu.edu.sa (F.B.); saalsalamah@imamu.edu.sa (S.A.A.); mialghonaim@imamu.edu.sa (M.I.A.); 5Department of Chemistry and Technologies of Drug, Sapienza University, P. le Aldo Moro, 5, 00185 Rome, Italy; stefania.garzoli@uniroma1.it

**Keywords:** *Dianthus sylvestris* subsp. *aristidis*, phytochemical profiling, LC-ESI-MS/MS, enzymatic inhibitory activities, photoprotective potential, secondary metabolites

## Abstract

**Background/Objectives**: This study provides the first comprehensive phytochemical composition and biological evaluation of *Dianthus sylvestris* subsp. *aristidis* (Batt.) Greuter & Burdet, a plant endemic to Algeria with unexplored pharmacological potential. The objective is to identify novel bioactive metabolites in the plant’s extracts and assess their potential applications for skincare and metabolic disorder management, addressing gaps in the current understanding of its medicinal value. **Methods**: Liquid chromatography coupled with electrospray ionization tandem mass spectrometry (LC-ESI-MS/MS) profiling was used to analyze the hydromethanolic (HMeOH) leaf extract and identify bioactive compounds. The biological activities of HMeOH, ethyl acetate (EtOAc), and butanolic (n-BuOH) extracts were tested for cytotoxicity using the brine shrimp lethality test, photoprotective potential by calculating the sun protection factor (SPF), and enzymatic inhibitory activities against alpha-amylase, urease, and tyrosinase. **Results**: The LC-ESI-MS/MS profiling of the MeOH extract identified 22 bioactive compounds, including phenolic acids and flavonoids, some of which have not been previously reported in this species. Cytotoxicity tests showed that all extracts were non-toxic (half-lethal concentration (LC_50_) > 100 micrograms per milliliter). The SPF values indicated significant photoprotective potential, with EtOAc (SPF = 45.19 ± 0.73) and n-BuOH (SPF = 43.81 ± 0.59) extracts showing high sun protection activity. The n-BuOH extract exhibited strong alpha-amylase inhibitory activity (half-maximal inhibitory concentration (IC_50_) = 307.08 micrograms per milliliter), surpassing the standard acarbose (IC_50_ = 3650.93 micrograms per milliliter), suggesting potential applications in diabetes management. **Conclusions**: *Dianthus sylvestris* subsp. *aristidis* demonstrates significant pharmacological potential as a source of bioactive secondary metabolites for skincare and metabolic disorder management. These findings provide new insights into the plant’s therapeutic potential and set a foundation for future pharmacological and clinical investigations.

## 1. Introduction

Phytomedicines have been essential in healthcare for centuries, particularly in regions where traditional medicine remains prevalent. Rooted in ancient civilizations, their therapeutic potential continues to be recognized globally. While modern pharmaceuticals have evolved from these traditional practices, herbal remedies persist due to their abundance of bioactive compounds, which remain essential in therapeutic fields like phytotherapy and aromatherapy [1]. The increasing demand for natural therapeutics has prompted further exploration of plant-derived bioactive compounds, particularly from endemic species with unique phytochemical profiles.

Medicinal plants exert their pharmacological effects through a diverse array of bioactive compounds, including phenolics, flavonoids, alkaloids, and terpenoids, which contribute to their cytotoxic, anti-inflammatory, antimicrobial, antioxidant, and enzyme inhibitory properties [2,3]. Endemic plants, in particular, are valuable resources due to their adaptation to specific ecological niches, often leading to the biosynthesis of unique secondary metabolites with potential therapeutic applications [4]. Algeria, a biodiversity-rich country within the Mediterranean region, harbors numerous endemic plants with untapped pharmacological potential. However, despite its high species diversity, many species remain unexplored, necessitating further phytochemical and biological studies.

One such under-researched endemic species is *Dianthus sylvestris* subsp. *aristidis* (Batt.) Greuter & Burdet, a rare subspecies of the *Dianthus* genus [5]. The taxonomic representation of *Dianthus* in Algeria has evolved over time; Quezel and Santa (1962) [6] reported seven taxa, whereas Dobignard and Chatelain (2011) [7] later revised this number to twelve after considering the extinction of *Dianthus tripunctatus* Sm.

While the pharmacological properties of several *Dianthus* species have been extensively studied demonstrating antibacterial, antifungal, antiviral, antioxidant, cytotoxic, antidiabetic, anticancer, anti-inflammatory, insecticidal, analgesic, anesthetic, renal-protective, and hepatoprotective activities [8,9,10,11,12,13,14,15], *Dianthus sylvestris* subsp. *aristidis* remains largely unexplored in terms of its phytochemical composition and bioactivity.

Beyond their medicinal applications, *Dianthus* species hold significance in the food and cosmetic industries. Edible *Dianthus* flowers are valued for their sensory attributes, including color, taste, and aroma, while also serving as natural sources of biologically active substances with antioxidant and anti-inflammatory properties [16,17]. Recent advancements in food-processing technologies, such as electron beam and gamma irradiation, have demonstrated that these methods can preserve the integrity and bioactivity of *Dianthus* flowers, making them promising ingredients in functional foods and nutraceuticals [18].

Recent research by Bouzana et al. (2023, 2025) [19,20] identified strong antioxidant, antibacterial, and antifungal activities in *Dianthus sylvestris* subsp. *aristidis.* However, its phytochemical composition, cytotoxicity, photoprotective effects, and enzyme-inhibitory potential remain unexplored. Investigating this subspecies could reveal novel applications in pharmaceutical, cosmetic, and food industries.

To address this gap, this study aims to analyze the phytochemical composition of *Dianthus sylvestris* subsp. *aristidis* leaf extracts using LC-MS/MS, evaluate their cytotoxicity to determine potential biomedical applications, assess their photoprotective effects through SPF analysis for potential use in skincare formulations, and investigate their enzyme-inhibitory activities against α-amylase, urease, and tyrosinase, which are relevant to diabetes management, food preservation, and cosmetics. While the specific bioactive compounds responsible for these activities remain to be identified, this work provides a foundation for the further exploration of the plant’s pharmacological potential.

## 2. Results and Discussion

### 2.1. Phytochemical Composition of HMeOH Extract: LC-MS/MS Results

Using LC-MS/MS, we identified 22 bioactive compounds in the HMeOH extract of *D. sylvestris* subsp. *aristidis* (Table 1, Figure 1), including phenolic acids (ferulic acid, chlorogenic acid, sinapic acid, caffeic acid, p-coumaric acid, and vanillic acid) and flavonoids (rutin, luteolin, epicatechin, naringenin, and chrysin). These compounds exhibit significant biological activities such as antioxidant properties that neutralize free radicals and reduce oxidative stress, potentially preventing chronic diseases like cancer and cardiovascular disorders. They also possess anti-inflammatory and antimicrobial activities, making them potential candidates for treating infections and inflammatory conditions. Furthermore, their cardiovascular benefits are evident in their ability to improve endothelial function and regulate blood pressure [21,22,23,24,25]. In addition to these polyphenolic compounds, the extract contains essential vitamins (riboflavin and folic acid) and other bioactive metabolites (beta-carotene, oleanolic acid, curcumin, resveratrol, thymol, and oleuropein). Notably, folic acid plays a critical role in DNA synthesis and repair [26]. Phytochemical studies on the *Dianthus* genus have identified a wide range of bioactive compounds, including, triterpenes, phenolics, flavonoids, anthocyanins, alkaloids, cyanogenic glucosides, coumarins, ecdysteroids, and essential oils [15]. Among them, rutin, phloridzin, ferulic acid, and chlorogenic acid are dominant in endemic species [12]. A comprehensive review on *Dianthus superbus* and *Dianthus chinensis* has reported 194 compounds, primarily saponins, peptides, anthraquinones, phenolic acids, amides, and phenylpropanoids [27]. *Dianthus caryophyllus* is rich in methyl ferulate, luteolin-4′-O-glucoside, anthraquinone, rhein-8-O-glucoside, isoorientin 2-O-rhamnoside, and kurarinone, known for their antioxidant properties [28]. Additionally, 6-hydroxykaempferol-3,6-O-diglucoside, 6-hydroxykaempferol-7-O-glucoside, quercetin-3-O-sophoroside, and 2′-deoxyguanosine contribute to its antioxidant and anticancer activities [18].

A comparison with the EtOAc and n-BuOH fractions of the HMeOH extract [19] highlighted the common presence of p-coumaric acid, chlorogenic acid, and folic acid, highlighting the similarity in phytochemical composition among different fractions. This consistency in compound presence is further supported by a comparative analysis with the aqueous extract of *Dianthus carmelitarum*, which revealed a significant overlap in phenolic compounds, including ferulic acid, chlorogenic acid, and p-coumaric acid [29]. Similarly, the isolation of trans-p-coumaric acid from the ethyl acetate fraction of *Dianthus superbus* ethanol extract [30] supports the shared presence of these compounds across related species. However, our analysis also detected additional bioactive compounds such as curcumin, resveratrol, thymol, and 4 methoxybenzoic acid that have not been previously reported in the context of *Dianthus* species. These findings underscore the novelty of our study by expanding the known phytochemical profile of *D. sylvestris* subsp. *aristidis* and enriching the existing literature with new insights into its potential therapeutic applications. The differences in phenolic composition observed may result from variations in the analytical standards used and the selective extraction properties of the solvents, as the polarity of the extraction solvent plays an essential role in determining which compounds are recovered.

To fully exploit the therapeutic potential of these compounds, future research should focus on their mechanisms of action, bioavailability, and potential synergistic interactions. Such studies will enhance our understanding of how to optimize the use of these compounds in medical applications.

### 2.2. Brine Shrimp Cytotoxicity

The cytotoxicity test on *Artemia salina* (*A. salina*) is a widely used bioassay due to its rapid and cost-effective nature, making it ideal for evaluating the toxicity of drugs, including plant extracts. The test is based on the concept that a substance is considered cytotoxic if its LC_50_ value (the concentration required to kill 50% of the test population) is lower than 100 µg/mL [31,32]. In this study, the cytotoxic activity of various extracts (HMeOH, EtOAc, and n-BuOH) of *D. sylvestris* subsp. *aristidis* was determined. As presented in Table 2, the LC_50_ values indicated that none of the extracts demonstrated cytotoxic effects on *A. salina* nauplii with values of 6320, 2500, and 1272 µg/mL, respectively.

The non-toxic profile of the extracts aligns with their phytochemical composition, which includes compounds with well-documented cytoprotective properties. For instance, the presence of phenolic acids (ferulic acid, chlorogenic acid, p-coumaric acid), as well as flavonoids (rutin, luteolin, epicatechin) in the HMeOH extract, likely contributes to the extract’s safety. Chlorogenic acid, for example, exhibits antioxidant activity by scavenging free radicals and modulating oxidative stress pathways, thereby protecting cells from damage [33]. Similarly, rutin and luteolin are known to stabilize cell membranes and reduce lipid peroxidation, mechanisms that mitigate cytotoxicity [34,35]. Notably, oleanolic acid and curcumin, also detected in this extract, have demonstrated hepatoprotective and anti-inflammatory effects in various preclinical models, further supporting the safety profile [36].

EtOAc and n-BuOH extracts contain flavonoids such as naringenin, quercetin, and kaempferol [20], all of which exhibit minimal cytotoxicity and are classified as low-risk substances, with LD_50_ values greater than 2000 mg/kg in acute toxicity tests on Wistar rats [37]. The low cytotoxicity of *D. sylvestris* subsp. *aristidis* contrasts sharply with reports on other *Dianthus* species. For example, *Dianthus basuticus* extracts exhibited LC_50_ values of 17.3–59.4 µg/mL [38]. This may be due to the presence of more cytotoxic saponins or alkaloids in *Dianthus basuticus*, which were not detected in *D. sylvestris* subsp. *aristidis.*

The safety profile of *D. sylvestris* subsp. *aristidis* extracts supports their potential for incorporation into medical and pharmaceutical applications and underscores their promise for dual use in skincare formulations and as natural additives in food preservation.

### 2.3. Sun Protection Factor Activity (SPF)

The sun protection factor (SPF) is the most commonly used parameter to measure the sun protection capacity of sunscreens [39]. The findings demonstrated that the SPF values for the different *D. sylvestris* subsp. *aristidis* extracts ranged from 17.82 ± 0.45 to 45.19 ± 0.73, as shown in Table 3 and Figure 2. In accordance with the Commission of European Communities (2006) [40], the EtOAc and n-BuOH extracts belong to the category of high protection activity (30–49.9), the same category of Reference 1 (Venus) and Reference 2 (Uriage). This positions these extracts as promising candidates for natural sunscreen formulations.

In fact, extended exposure to ultraviolet (UV) radiation, especially UVA and UVB rays, can damage the face, neck, head, back of the hands, and other frequently sun exposed areas of the body [41]. Under certain conditions, this can lead to several harmful consequences, such as sunburn, photocarcinogenesis, immunosuppression, early aging, and skin cancer [42]. To prevent these detrimental effects, many natural sun care products, such as plant extracts, are used. Plants have evolved a number of defense mechanisms against UV radiation, including the ability to filter UV wavelengths and repair UV-induced damage [43].

The high SPF values observed in the EtOAc and n-BuOH extracts suggest that these fractions contain bioactive molecules capable of mitigating UV-induced damage through multiple mechanisms.

The phenolic acids present in extracts, including chlorogenic, ferulic, and p-coumaric acid, further enhance SPF activity by stabilizing free radicals and protecting skin cells from lipid peroxidation. Ferulic acid, in particular, has been shown to increase the photostability of sunscreen formulations when combined with other UV filters, suggesting its potential application in natural photoprotective formulations [44].

Flavonoids such as quercetin, naringenin, and kaempferol, which are abundant in extracts [20], contribute to photoprotection by absorbing UVA and UVB radiation due to their conjugated aromatic systems. Additionally, these compounds act as powerful antioxidants, neutralizing reactive oxygen species (ROS) generated by UV exposure and preventing oxidative stress, a major contributor to photoaging and skin damage [45,46].

Carotenoids such as β-carotene, detected in the EtOAc and n-BuOH extracts [20], provide additional protection by quenching singlet oxygen species and reducing erythema and DNA damage. In addition, the presence of ascorbic acid and riboflavin adds to the photoprotective properties by promoting antioxidant defense and DNA repair mechanisms [39,47].

The synergy between these bioactive molecules explains the significant SPF values recorded in this study, aligning with the photoprotective potential of plant-derived compounds. However, while our in vitro results strongly support the photoprotective potential of these extracts, in vivo confirmation remains essential to validate their efficacy, photostability, and skin penetration under real-world conditions. Future studies should focus on these aspects to further establish *D. sylvestris* subsp. *aristidis* as a viable source of natural photoprotective agents.

### 2.4. Enzymatic Inhibitory Activities

#### 2.4.1. Alpha-Amylase Inhibitory Activity

The α-amylase inhibitory findings are represented in Table 4 and Figure 3. The n-BuOH extract with (IC_50_ = 307.08 ± 1.13 µg/mL) was better than the EtOAc extract, which was inactive at 400 µg/mL, and failed to reach the 50% of the enzyme inhibition level, while the HMeOH extract exerted no α-amylase inhibitory effect. The n-BuOH extract was also better than the acarbose standard (IC_50_ = 3650.93 ± 10.70 µg/mL), demonstrating nearly 11-fold higher inhibitory activity. However, it was less effective than the ethanolic extract of *Dianthus basuticus*, which displayed significantly higher inhibition (IC_50_ = 34.02 mg/mL) [10].

Diabetes is a chronic epidemic disease that develops when insulin synthesis is insufficient, its secretion is impaired, or its binding to its receptor (IR) on the cell is ineffective, leading to a significant increase in blood glucose levels, thereby damaging the body system [48]. The key enzyme present in saliva and the pancreas, catalyzing the cleavage of starches, is α-amylase [49]. This enzyme cleaves the glycosidic linkages in α-D-(1,4) carbohydrates to produce oligosaccharides, which alpha-glucosidase then breaks down into monosaccharide, facilitating their absorption and leading to hyperglycemia [49,50]. The inhibition of α-amylase activity prevents the breakdown and absorption of complex carbohydrates, thereby reducing the risk of developing type 2 diabetes [51].

Synthetic α-amylase inhibitors, such as acarbose, are widely used in diabetes treatment, but often induce gastrointestinal adverse effects that include ulcerations, diarrhea, hernias, and bloating in the abdomen [52]. Some natural forms of α-amylase, such as phenolic compounds and flavonoids derived from plants, can be employed in therapy to prevent and treat type 2 diabetes with few adverse effects [52,53]. This involves inhibiting complex carbohydrate breakdown and absorption to lower postprandial blood glucose levels [54]. Moreover, certain plant-derived minerals enhance insulin activity, further contributing to glycemic control [55].

This study demonstrated that the n-BuOH extract of *D. sylvestris* subsp. *aristidis* has good in vitro anti-diabetic potential by inhibiting α-amylase. This inhibition is likely attributed to the presence of phenolic compounds: gallic acid, p-coumaric acid, chlorogenic acid, naringin, hesperetin, and kaempferol [56], previously reported in the phytochemical profile of this extract [19]. These compounds are known to inhibit α-amylase through multiple interactions with the enzyme’s catalytic site for their ability to interact with the enzyme’s active site.

Polyphenols exert their inhibitory effect through hydrogen bonding between their hydroxyl groups and the catalytic residues of α-amylase (e.g., Asp197 and Glu233) as well as hydrophobic interactions with aromatic residues like tryptophan (Trp59), reducing enzymatic activity [57].

Flavonoids, in particular, inhibit α-amylase by forming stable conjugated π-π interactions between their benzopyrone rings and the indole ring of Trp59, interfering with substrate binding. Structural elements, such as the presence of a C2–C3 double bond, hydroxylation patterns on rings A and B, and additional hydroxyl groups at key positions, significantly influence their inhibitory potency [58].

Hydroxycinnamic acids contribute to enzyme inhibition through a conjugated system that facilitates electron transport and enhances hydrophobic interactions with the enzyme’s active site. In contrast, hydroxybenzoic acids like salicylic and vanillic acid have shown weak or no inhibitory effects [59].

These findings suggest that the α-amylase inhibitory activity of *D. sylvestris* subsp. *aristidis* may be due to a combination of these molecular interactions, warranting further investigation through molecular docking and enzyme kinetics studies to elucidate precise binding mechanisms.

#### 2.4.2. Urease Inhibitory Activity

Urease is an enzyme responsible for breaking down urea into ammonia and carbon dioxide through hydrolysis, a process that is crucial for several organisms, including *Helicobacter pylori*, which uses urease to neutralize stomach acidity and potentially cause gastrointestinal issues like gastric ulcers [60,61].

The urease inhibitory activity of the three extracts from *D. sylvestris* subsp. *aristidis* was evaluated, but no significant inhibitory effect was observed (Table 5). This lack of activity could be attributed to several factors.

One possible explanation is the absence or low concentration of key phenolic compounds known for their urease inhibitory properties.

Several potent inhibitors, such as catechin, epigallocatechin gallate, methyl gallate, procyanidins, quercetin, myricetin, luteolin-7-O-glucuronide, 5,7-dihydroxyflavone, and isoflavonoids like genistein, ponciretin, and baicalin, have been identified in previous studies but may not be present in significant amounts in these extracts [62,63,64,65].

Additionally, while some weaker urease inhibitors such as resveratrol, naringin, epicatechins, and luteolin were detected, their concentrations might have been too low to exert a measurable effect. Another possibility is that the solvent polarity used during extraction was not optimal for isolating the most effective urease-inhibiting compounds.

To address these limitations, future studies should explore alternative extraction methods using different solvents to enhance the recovery of active urease inhibitors. Further quantitative analysis of specific bioactive compounds could also help clarify their contribution to urease inhibition.

#### 2.4.3. Tyrosinase Inhibitory Activity

Tyrosinase is an enzyme found in various organisms, including edible mushrooms [66]. It plays a crucial role in melanin synthesis, the pigment responsible for the color of the skin, hair, and eyes in humans and animals [67]. Melanin is important to protect the skin from damage caused by ultraviolet (UV) radiation. However, excessive production of tyrosinase leads to an overproduction of this pigment, resulting in various dermatological disorders, particularly pigmentation disorders. Therefore, inhibiting tyrosinase is a target for regulating melanin production [68] and treating conditions such as hyperpigmentation, acne, and lentigines [69], which helps reduce dark spots and even out skin pigmentation. Synthetic tyrosinase inhibitors are limited due to their poor solubility and toxicity. For this reason, studies have focused on developing natural inhibitors with lower toxicity, such as polyphenols, flavonoids, essential oils, and alkaloids [70].

The tyrosinase inhibitory activity of the three extracts was studied, but none of the extracts exhibited significant inhibition (Table 6). This could be explained by the absence or low concentration of potent tyrosinase inhibitors such as flavonoids (e.g., kaempferol, quercetin, morin, catechin, rhamnetin) and stilbenes (e.g., resveratrol, piceatannol, oxyresveratrol, chlorophorin, and andalasin), which have been reported as effective tyrosinase inhibitors [71,72,73].

The extraction process and solvent polarity may have also influenced the presence of these bioactive compounds in the tested extracts. Further studies should focus on optimizing extraction conditions or screening other plant fractions for potential tyrosinase inhibitory activity.

## 3. Materials and Methods

### 3.1. Plant Material and Extraction

*D. sylvestris* subsp. *aristidis* (Figure 4) leaves were harvested from their natural habitat in the Îlot des Chèvres locality (36.881456, 6.927103) in Skikda, northeastern Algeria, in 2020. The botanical identification was conducted by Dr. Sakhraoui Nora from the Department of Ecology and Environment, University of 20 August 1955, Skikda, Algeria. Following collection, the leaves were cleaned and left to air-dry for 15 days before being finely ground into powder. Leaf extracts using HMeOH, EtOAc, and n-BuOH were prepared following the method described by Upson et al. (2000) [74]. A total of 100 g of powdered plant leaf material was macerated in 3 L of methanol 70% over three days with 1 L added daily, while being continuously stirred magnetically in a dark environment at ambient temperature. The mixture was then filtered through Whatman No. 1 filter paper, and the filtrate was concentrated under reduced pressure at 50 °C using a rotary evaporator to obtain the hydro-methanolic extract. A portion of the result extract was dissolved in 100 mL of distilled water and underwent liquid–liquid extraction in a separatory funnel with solvents: hexane (100 mL × 3) to eliminate fats and waxes, followed by ethyl acetate (100 mL × 3), and n-butanol (100 mL × 3). The solvents were subsequently evaporated under reduced pressure at 50 °C.

### 3.2. Phytochemical Profile of HMeOH Extract via LC-ESI-MS/MS

The phytochemical composition of the HMeOH extract was analyzed qualitatively at the Technical Platform of Physico-Chemical Analysis (PTAPC-CRAPC) in Ouargla, Algeria, using an ultra-high sensitivity UPLC-ESI-MS-MS (Shimadzu Kyoto, Japan) 8040 equipped with UFMS technology and a Nexera XR LC-20AD binary pump. Separation was achieved using an Ultra-force C18 column (dimensions: 150 mm × 4.6 mm, particle size: 3 μm; Restek, Bellefonte, PA, USA). Chromatographic separation was performed using water with 0.1% formic acid as mobile phase A and methanol as mobile phase B. The following gradient elution program was applied: A (98%) 0 min to 0.2 min, A (25%) 0.2 min to 7.5 min, A (0%) 7.5 min to 12.5min, A (0%) 12.5 min to 17 min, A (98%) 17 min to 18min, A (98%) 18 min to 21 min. The flow rate was 0.2 mL/min, the injection volume was 5 μL, and the column temperature was set to 30 °C. The ESI conditions for electro spray ionization were as follows: 230 KPs of CID gas; −6.00 kV conversion dynode; 350 °C interface temperature; 250 °C DL temperature; 3.00 L/min nebulization gas flow; 400 °C heat block temperature; and 10.00 L/min drying gas flow [19].

### 3.3. Brine Shrimp Lethality Bioassay

The brine shrimp lethality bioassay was conducted following the protocol of Meyer et al. (1982) [31]. *A. salina* eggs were hatched in a 22 × 32 cm rectangular plastic dish filled with seawater, divided by a perforated partition into a darkened main compartment and a lighted secondary compartment, where approximately 50 mg of shrimp eggs were dispersed.

For the bioassay, ten nauplii were placed in each vial containing 0.5 mL of tested extract and 4.5 mL of seawater. The same conditions were maintained without the tested extracts, serving as a negative control, while the potassium dichromate was used as a positive control. After 24 h, the surviving shrimps were counted.

The lethal concentration (LC_50_) representing the concentration required to kill 50% of the tested population was determined using Probit Analysis and linear regression [76]. The LC_50_ was calculated in R (version 4.1.1.) using ‘ecotox’ package [77].

### 3.4. Sun Protection Factor (SPF)

The sun protection factor activity of the various extracts HMeOH, EtOAc, and n-BuOH was evaluated according to the method described by Mansur et al. (1986) [78]. The absorbance was measured between 290 and 320 nm at five distinct wavelengths. The following equation was used to calculate the SPF. Each measurement was made three times. Additionally, to compare the effectiveness of the extracts with commercial formulations, the SPF values of Venus and Uriage sunscreens were also determined.$${\mathrm{SPF}}_{\mathrm{spectrophotometric}}=\mathrm{CF}\times \sum_{290}^{320}EE\left(\lambda \right)\times I\left(\lambda \right)\times Abs(\lambda )$$

*EE*: erythemal effect spectrum, *I*: solar intensity spectrum, *Abs*: sunscreen product’s absorption, *CF*: the correction factor (=10), *EE* ∗ *I*: constant values established by Sayre et al. (1979) [79], as shown in (Table 7).

### 3.5. Inhibition of Enzymatic Activities

#### 3.5.1. Inhibition of Alpha-Amylase Activity

The α-amylase inhibitory activity was carried out using iodine/potassium iodide (IKI) technique as described by Zengin et al. (2014) [80] with slide modifications. A mixture of 50 µL of α-amylase solution (1U) prepared in phosphate buffer (pH 6.9 with 6 mM sodium chloride) and a volume of 25 uL of extracts solutions (HMeOH, EtOAc, and n-BuOH) at different concentrations (400; 200; 100; 50; 25; 12.5; and 6.25 µg/mL) was incubated for 10 min at 37 °C. Afterward, 50 µL of a 0.1% starch solution was added, and the mixture was re-incubated for 10min at 37 °C. After incubation, 25 µL of HCl (1M) and 100 µL of iodine potassium iodide (IKI) were added. Using a 96-well microplate reader (PerkinElmer, EnSpire Multimode Plate Reader, Waltham, MA, USA), the absorbance was read at 630 nm. Acarbose was used as a standard. The inhibition percentage of α-amylase was calculated using the following equation.
α-amylase inhibition % = 1 − [(A_c_ − A_e_) − (A_s_ −A_b_)/(A_c_ − A_e_)]
A_c_ = Absorbance [Stratch + IKI + HCl + Solvent vol extract + Enzyme sodium phosphate buffer]; A_e_ = Absorbance [Enzyme + Stratch + IKI + HCL + Solvent vol Extract]; A_s_ = Absorbance [Enzyme + Extract + Strach + IKI + HCl]; A_b_ = Absorbance [Extract + IKI + Sodium phosphate buffer].

#### 3.5.2. Inhibition of Urease Activity

The urease inhibitory activity was assessed using the indophenol method to measure ammonia generation as detailed by Weatherburn (1967) [81]. The reaction mixture comprising 10 μL of each extract (HMeOH, EtOAc, and n-BuOH), 25 μL of the enzyme solution (Jack bean urease), and 50 μL of urea solution was placed in 96-well plate and incubated at 30 °C for 15 min. Each well was then filled with 45 μL of phenol reagent and 70 μL of alkaline reagent. The absorbance was measured at 630 nm using a 96-well microplate reader (PerkinElmer, EnSpire Multimode Plate Reader, Walthan, MA, USA) following a 50 min incubation period. The standard inhibitor was thiourea. The inhibition percentage of urease was calculated using the following equation:Inhibition (%) = [(A control − A extract)/(A control)] × 100
where A control is the absorbance of the negative control, and A extract is the absorbance of the extract/standard.

#### 3.5.3. Inhibition of Tyrosinase Activity

Tyrosinase was extracted using the method of Gouzi and Benmansour (2007) [82]. A total of 100 g of *Agaricus bisporus* (button mushrooms) purchased from a local market was blended with 120 mL of pre-cooled phosphate buffer pH 7 for 30 s. After 30 min of agitation and filtration, the filtrate was centrifuged at 18,000× *g* for 30 min at 4 °C, yielding the crude enzymatic extract of tyrosinase. The tyrosinase inhibitory activity was determined using the method of Deveci et al. (2018) [83], where 10 µL of each extract (HMeOH, EtOAc, and n-BuOH) was combined with 150 µL of phosphate buffer (pH 6.8), following by the addition of 20 µL of tyrosinase enzyme and incubation at 37 °C for 10 min. Afterward, 20 µL of levodopa (L-DOPA) was added, and the mixture was incubated for another 10 min at 37 °C. The absorbance of the extracts against a blank was read at 475 nm using a 96-well microplate reader (PerkinElmer, EnSpire Multimode Plate Reader). The standard utilized was kojic acid. The inhibition percentage of tyrosinase was calculated using the following equation:Inhibition (%) = [(A control − A extract)/(A control)] × 100
where A control is the absorbance of the negative control and A extract is the absorbance of the extract/standard.

### 3.6. Statistical Analysis

Results were expressed as the mean value ± SD of the three measurements. IBM SPSS Statistics, version 25, was used to analyze the data and detect significant differences at *p* < 0.05 using one-way ANOVA variance and Tukey’s test.

## 4. Conclusions

This study demonstrated that *Dianthus sylvestris* subsp. *aristidis* leaves are a rich source of diverse bioactive compounds, with the hydromethanolic extract revealing a complex phytochemical profile dominated by phenolic acids and flavonoids. Notably, none of the three extracts exhibited cytotoxic effects (LC_50_ > 100 µg/mL), suggesting their potential safety. However, as this is a preliminary toxicity test, further studies on human cell lines are essential to confirm the safety of *D. sylvestris* subsp. *aristidis* extracts for human applications. The n-BuOH extract, in particular, showed strong photoprotective potential along with potent α-amylase inhibitory activity, exceeding that of the standard acarbose, highlighting its potential for managing postprandial hyperglycemia and, by extension, diabetes mellitus. These findings suggest promising applications in skincare formulations and natural food preservatives, addressing key challenges in the cosmetic and food sectors. However, the rarity of *D. sylvestris* subsp. *aristidis* in northeastern Algeria posed a limitation, as only small populations exist in rugged difficult-to-access environments. To prevent ecological harm, limited quantities were collected, restricting the scope of biological testing. Additionally, the absence of in vivo validation and long-term stability assessments remains a limitation, and the precise molecular mechanisms underlying the observed bioactivities require further investigation. Future research should focus on in vivo pharmacological validation, pharmacokinetics, the stability of formulated products, and potential synergistic interactions between bioactive compounds to further explore the plant’s applications in pharmaceuticals, cosmetics, and functional foods.

## Figures and Tables

**Figure 1 pharmaceuticals-18-00578-f001:**
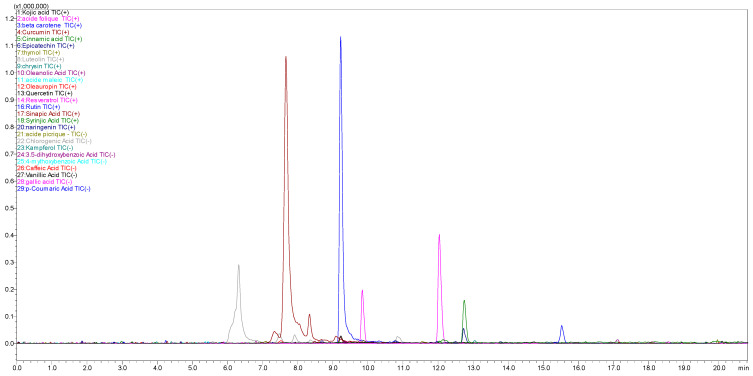
LC–MS/MS chromatogram of HMeOH extract of *D. sylvestris* subsp. *aristidis*.

**Figure 2 pharmaceuticals-18-00578-f002:**
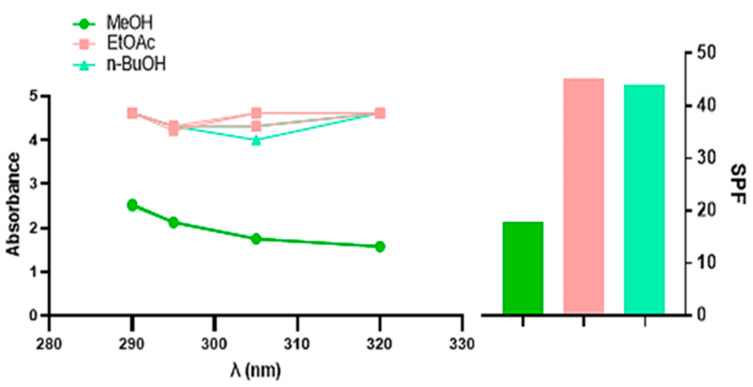
The SPF values of different *D. sylvestris* subsp. *aristidis* extracts.

**Figure 3 pharmaceuticals-18-00578-f003:**
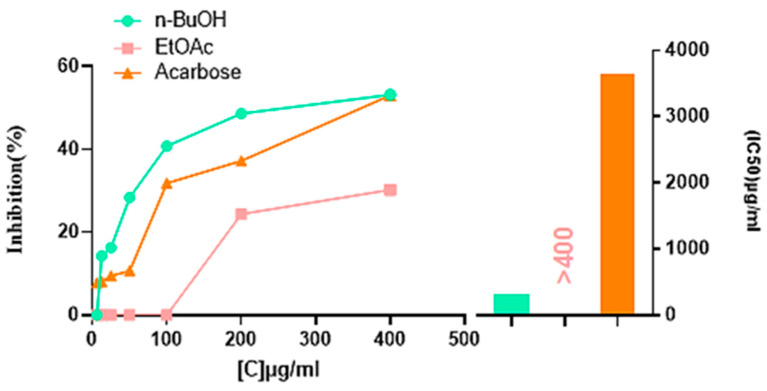
The α-amylase inhibitory activity of different *D. sylvestris* subsp. *aristidis* extracts.

**Figure 4 pharmaceuticals-18-00578-f004:**
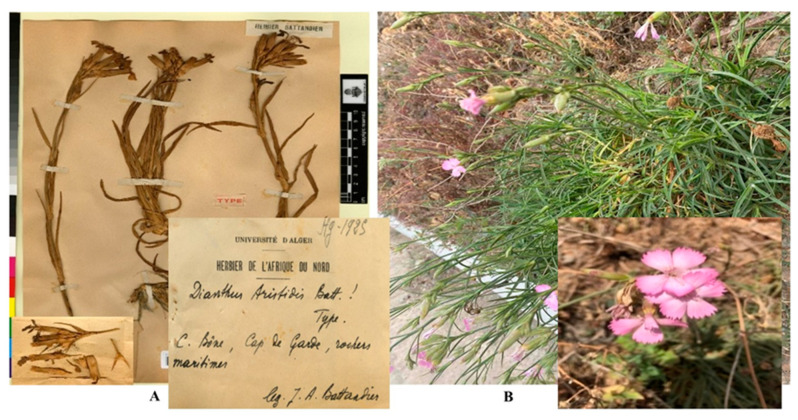
*D. sylvestris* subsp. *aristidis* (Batt.) Greuter et Burdet. (**A**) Herbarium sheets at Cap de Garde, Annaba (GBIF, 2023) [75]. (**B**) Photograph at Îlot des Chèvres, Skikda.

**Table 1 pharmaceuticals-18-00578-t001:** LC-MS/MS chemical profile of HMeOH extract of *D. sylvestris* subsp. *aristidis*.

ID	Name	Molecular Formula	Molecular Weight	ESI Charge (+/−)	*m*/*z*	Ret. Time	Height	Area
1	Ferulic acid	C_10_H_10_O_4_	194.18	(+)	194.8000 > 177.0500	8.610	3,736,345	19,847,102
2	Chlorogenic Acid	C_16_H_18_O_9_	354.31	(−)	353.0500 > 191.1000	6.323	257,094	2,536,296
3	Sinapic Acid	C_11_H_12_O_5_	224.21	(+)	225.0000 > 91.1000	9.215	13,522	63,556
4	Caffeic Acid	C_9_H_8_O_4_	180.16	(−)	179.1500 > 135.0000	7.521	8033	53,166
5	Cinnamic acid	C_9_H_8_O_2_	148.16	(+)	149.0500 > 117.0000	12.141	11,812	56,679
6	Syringic Acid	C_9_H_10_O_5_	198.17	(+)	199.0000 > 155.1500	20.210	5779	12,301
7	p-Coumaric Acid	C_9_H_8_O_3_	164.16	(−)	163.0500 > 118.9500	8.660	3291	9547
8	Vanillic Acid	C_8_H_8_O_4_	168.15	(−)	166.9500 > 151.9000	7.372	966	5209
9	Rutin	C_27_H_30_O_16_	610.5	(+)	611.0000 > 465.2000	9.226	208,513	1,157,023
10	Luteolin	C_15_H_10_O_6_	286.24	(+)	286.7500 > 153.0000	10.844	11,315	85,663
11	Epicatechin	C_15_H_14_O_6_	290.27	(+)	290.9000 > 123.1000	11.966	7332	36,849
12	Naringenin	C_15_H_12_O_5_	272.25	(+)	273.0500 > 153.0000	10.781	5456	35,475
13	Chrysin	C_15_H_10_O_4_	254.24	(+)	255.1000 > 68.8500	11.332	1667	6144
14	β carotene	C_40_H_56_	536.87	(+)	537.2000 > 23.1000	15.521	63,553	393,908
15	Oleanolic Acid	C_30_H_48_O_3_	456.7	(+)	457.3000 > 411.5000	17.117	9012	47,730
16	Riboflavin	C_17_H_20_N_4_O_6_	376.4	(+)	377.9000 > 361.3500	14.721	4,026,753	25,896,378
17	Folic Acid	C_19_H_19_N_7_O_6_	441	(+)	442.9000 > 323.4500	18.539	4865	31,602
18	Curcumin	C_21_H_20_O_6_	368.4	(+)	368.9000 > 145.0500	7.672	148,455	1,135,080
19	Resveratrol	C_14_H_12_O_3_	228.24	(+)	229.0500 > 135.1000	9.842	115,557	575,817
20	Thymol	C_10_H_14_O	150.22	(+)	151.1000 > 109.0500	11.547	3160	13,242
21	4-Methoxybenzoic Acid	C_8_H_8_O_3_	152.15	(+)	151.0500 > 107.0500	8.348	2403	10,040
22	Oleuropein	C_25_H_32_O_13_	540.5	(−)	541.3000 > 524.5500	19.765	1244	7789

**Table 2 pharmaceuticals-18-00578-t002:** The cytotoxic activity of different *D. sylvestris* subsp. *aristidis* extracts against brine shrimp nauplii.

Tested Extracts	Concentrations (µg/mL)	Initial Number of Nauplii	Total Death			Percentage of Letality	LC_50_ (µg/mL)	Confidence Interval (LCL-UCL)
HMeOH	1000	10	2	5	3	33.33	6320	1.18 × 10^3^–6.01 × 10^9^
	500	10	3	3	5	36.66		
	250	10	1	2	3	20		
	125	10	0	1	4	16.66		
	62.5	10	1	2	2	16.66		
	31.25	10	1	0	1	6.66		
EtOAc	1000	10	4	4	4	40	2500	6.75 × 10^2^–1.44 × 10^7^
	500	10	3	4	5	40		
	250	10	3	2	3	30		
	125	10	3	3	0	20		
	62.5	10	3	2	2	23.33		
	31.25	10	2	2	1	16.66		
n-BuOH	1000	10	5	5	6	53.33	1272	436–9.21 × 10^5^
	500	10	4	4	3	36.66		
	250	10	2	2	3	23.33		
	125	10	1	3	2	20		
	62.5	10	1	4	1	20		
	31.25	10	1	2	2	16.66		
Control	/	10	0	1	0	3.33	/	/

LCL: Lower confidence limit. UCL: Upper confidence limit. HMeOH: hydromethanolic extract; EtOAc: ethyl acetate extract; n-BuOH: butanolic extract

**Table 3 pharmaceuticals-18-00578-t003:** The SPF values of different *D. sylvestris* subsp. *aristidis* extracts.

Extracts	HMeOH	EtOAc	n-BuOH	Venus	Uriage
SPF	17.82 ± 0.45 ^a^	45.19 ± 0.73 ^b^	43.81 ± 0.59 ^b^	50.11 ± 0.53 ^c^	44.22 ± 0.3 ^b^

Values expressed as mean ± SD of three parallel measurements. Values followed by different letters (a–c) are significantly different (*p* < 0.05). SPF: Sun Protection Factor; HMeOH: hydromethanolic extract; EtOAc: ethyl acetate extract; n-BuOH: butanolic extract.

**Table 4 pharmaceuticals-18-00578-t004:** The α-amylase inhibitory activity of different *D. sylvestris* subsp. *aristidis* extracts.

Extracts	% Inhibition of α-Amylase							IC_50_ (µg/mL)
	**6.25 µg**	**12.5 µg**	**25 µg**	**50 µg**	**100 µg**	**200 µg**	**400 µg**	**IC_50_ (µg/mL)**
HMeOH	NA	NA	NA	NA	NA	NA	NA	NA
EtOAc	NA	NA	NA	NA	NA	24.40 ± 0.36	30.24 ± 1.74	˃400
n-BuOH	NA	14.27 ± 1.04	16.27 ± 2.87	28.36 ± 0.78	40.75 ± 0.79	48.64 ± 0.42	53.19 ± 0.04	307.08 ± 1.13 ^b^
	**62.5 µg**	**125 µg**	**250 µg**	**500 µg**	**1000 µg**	**2000 µg**	**4000 µg**	**IC_50_ (µg/mL)**
Acarbose	7.76 ± 0.17	8.08 ± 0.30	9.46 ± 0.11	10.70 ± 0.96	31.81 ± 2.89	37.21 ± 3.54	53.05 ± 1.59	3650.93 ± 10.70 ^a^

Values are expressed as means ± SD (n = 3). NA: no activity. Values followed by different letters (a,b) are significantly different (*p* < 0.05); HMeOH: hydromethanolic extract; EtOAc: ethyl acetate extract; n-BuOH: butanolic extract.

**Table 5 pharmaceuticals-18-00578-t005:** The urease inhibitory activity of different *D. sylvestris* subsp. *aristidis* extracts.

Extracts	IC_50_ µg/mL
HMeOH	NA
EtOAc	NA
n-BuOH	NA
Thiourea	11.57 ± 0.68

Values are expressed as means ± SD (n = 3). NA: no activity; HMeOH: hydromethanolic extract; EtOAc: ethyl acetate extract; n-BuOH: butanolic extract.

**Table 6 pharmaceuticals-18-00578-t006:** The tyrosinase inhibitory activity of different *Dianthus sylvestris* subsp. *aristidis* extracts.

Extracts	IC_50_ µg/mL
HMeOH	NA
EtOAc	NA
n-BuOH	NA
Kojic acid	19.43 ± 0.98

Values are expressed as means ± SD (n = 3). NA: no activity; HMeOH: hydromethanolic extract; EtOAc: ethyl acetate extract; n-BuOH: butanolic extract.

**Table 7 pharmaceuticals-18-00578-t007:** The standardized product function used to calculate the sun protection factor [78].

Wavelength λ (nm)	EE (λ) × I(λ) (Norms)
290	0.0150
295	0.0817
300	0.2874
305	0.3278
310	0.1864
315	0.0837
320	0.0180
Total	1.0000

## Data Availability

Data is contained within the article.

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
