# Peer review of "Phytochemical Profiling and Biological Evaluation of Dianthus sylvestris subsp. aristidis: A Chromatographic and Mass Spectrometry Approach to Uncovering Bioactive Metabolites for Dermatological and Metabolic Disorder Management"

_pharmaceuticals, 2025, doi:10.3390/ph18040578_

Round 1
Reviewer 1 Report
Comments and Suggestions for Authors
Although the article provides important findings on the phytochemical profile and biological activities of Dianthus sylvestris subsp. aristidis, it requires a comprehensive revision due to methodological and discussion deficiencies.
My section-by-section evaluations of the article are given below.
1. Title and Abstract
Title: It reflects the content of the study accurately. However, the "Skincare and Diabetes Management" section is too broad. A more specific statement could be used.
Abstract: Generally comprehensive. However, the innovative aspects of the study and how it contributes to the existing literature could be emphasized more clearly. In addition, a brief summary of the methods used would be useful.
2. Introduction
- The research gap should be defined more clearly.
More up-to-date sources on why endemic plants are important could be added.
- Hypothesis or research questions are not clearly stated.
3. Methods
- Phytochemical Profiling (LC-ESI-MS/MS): Detailed explanation of the methods is positive. However, the following deficiencies are noteworthy:
The verification methods (e.g., use of reference standards) of the analyzed compounds are not specified.
- Information on the reproducibility and validation of the data is lacking.
- Details on positive and negative controls are not provided.
- Biological Tests:
- Artemia salina cytotoxicity test is well described. However, it is recommended to include a positive control (e.g., a potential toxic agent) for cytotoxicity assessment.
- Comparison with commercial sunscreen formulations is lacking in the SPF test.
4. Results and Discussion
- Detailed description of the chemical components is positive. However, LC-MS/MS chromatograms or spectrograms are not visually presented.
- The relationship of some detected compounds with biological activities is not discussed in depth.
- α-Amylase inhibition part is strong, but further comments can be made on the inhibitory mechanism.
- Urease and tyrosinase inhibition results were negative, but the reasons for this situation are not discussed.
- In vivo confirmation of photoprotective activity is lacking.
5. Conclusions
- Limitations of the study are not stated.
- More detailed suggestions can be made for future studies.
Author Response
Reviewer 1 (CHANGES ARE MARKED IN YELLOW)
The corrections have been highlighted in the document by a yellow color
Quality of English Language
The English is fine and does not require any improvement.
|
Yes |
Can be improved |
Must be improved |
Not applicable |
|
|
Does the introduction provide sufficient background and include all relevant references? |
( ) |
(x) |
( ) |
( ) |
|
Is the research design appropriate? |
( ) |
(x) |
( ) |
( ) |
|
Are the methods adequately described? |
( ) |
(x) |
( ) |
( ) |
|
Are the results clearly presented? |
( ) |
(x) |
( ) |
( ) |
|
Are the conclusions supported by the results? |
( ) |
(x) |
( ) |
( ) |
By responding all comments, we have improved all requested sections in table
Comments and Suggestions for Authors
Although the article provides important findings on the phytochemical profile and biological activities of Dianthus sylvestris subsp. aristidis, it requires a comprehensive revision due to methodological and discussion deficiencies.
My section-by-section evaluations of the article are given below.
1.Title and Abstract
Title: It reflects the content of the study accurately. However, the "Skincare and Diabetes Management" section is too broad. A more specific statement could be used.
*You have right, we have change: Skincare and Diabetes Management" by: Dermatological and Metabolic Disorder Management; Please check the revised version; title
Abstract: Generally comprehensive. However, the innovative aspects of the study and how it contributes to the existing literature could be emphasized more clearly.
*Innovative aspects: we highlighted that this is the first comprehensive phytochemical and biological evaluation of Dianthus sylvestris subsp. aristidis.
*We clarified that some of the identified bioactive compounds have not been previously reported in this species.
* We emphasized how the findings provide new insights into its therapeutic potential and play a foundation for future pharmacological and clinical research, please check the abstract
In addition, a brief summary of the methods used would be useful.
* A brief summary of the methods used have been added, please check the abstract.
- Introduction
- The research gap should be defined more clearly.
* The revised introduction now explicitly states the research gap. The updated introduction clarifies why this subspecies remains unexplored and why further investigation is necessary. (Lines 63-65, 74,75, )
More up-to-date sources on why endemic plants are important could be added.
* Additional reference and source have been incorporated to emphasize the significance of endemic plants in pharmacological and industrial applications (Lines 56, 60-62, 493)
- Hypothesis or research questions are not clearly stated.
* The study hypothesis have been integrated within the introduction. (Lines 85-88) please check the introduction
- Methods
- Phytochemical Profiling (LC-ESI-MS/MS): Detailed explanation of the methods is positive. However, the following deficiencies are noteworthy:
The verification methods (e.g., use of reference standards) of the analyzed compounds are not specified.
We utilized a Shimadzu 8040 LC-MS/MS spectrometer for our analysis.
Optimization of fragment ions and voltages (Q1, CE, Q2) for each compound of interest was performed using pure standards of polyphenols.
These standards were directly injected into the mass spectrometer, bypassing the chromatographic column, to obtain optimal reference spectra for each compound.
This process enabled the creation of a reference MRM database (positive [MH+] or negative [MH-] MRM) for the studied polyphenols.
- Information on the reproducibility and validation of the data is lacking. And Details on positive and negative controls are not provided.
Our study focuses on the qualitative analysis of polyphenols in complex samples.
Due to the significant matrix effect observed in these samples, precise quantitative reproducibility is challenging to establish.
However, the reliability of compound identification is based on comparing the obtained MRM spectra with the reference spectra in our database.
The method was validated by the use of pure standards to create the database.
We have included that The phytochemical composition of the MeOH extract was analyzed qualitatively ( line 354)
- Biological Tests:
- Artemia salina cytotoxicity test is well described. However, it is recommended to include a positive control (e.g., a potential toxic agent) for cytotoxicity assessment.
* We have included details regarding the positive (potassium dichromate) and negative controls (lines 375, 376)
- Comparison with commercial sunscreen formulations is lacking in the SPF test.
* We have included a comparison between the SPF values of our extracts and those of commercial sunscreens Venus and Uriage (Lines 385- 387)
- Results and Discussion
- Detailed description of the chemical components is positive. However, LC-MS/MS chromatograms or spectrograms are not visually presented.
* The he LC-MS/MS chromatograms have been included (Figure 1) line 146
- The relationship of some detected compounds with biological activities is not discussed in depth.
* We have revised the manuscript to better discuss the biological relevance of the detected compounds.
Lines (cytotoxicity) 157-168, 200-214 (SPF), 256- 259 (alpha amylase)
- α-Amylase inhibition part is strong, but further comments can be made on the inhibitory mechanism.
* We have expanded the discussion on the α-amylase inhibitory mechanism by detailing the key interactions involved lines 261-277
- Urease and tyrosinase inhibition results were negative, but the reasons for this situation are not discussed.
* We highlighted the absence or low concentration of potent urease/ tyrosinase inhibitors and the presence of weaker inhibitors at potentially insufficient levels. We also considered the impact of solvent polarity on compound extraction. Lines 291-300(urease) 320-324 (tyrosinase)
- In vivo confirmation of photoprotective activity is lacking.
* We have incorporated the point on in vivo confirmation into the discussion, emphasizing the need for further validation of the extracts' photoprotective efficacy. Lines 215,221
- Conclusions
- Limitations of the study are not stated.
* We have addressed the study's limitations by highlighting the restricted availability of Dianthus sylvestris subsp. aristidis due to its rarity and inaccessible habitat, the absence of in vivo validation and long-term stability assessments of the extracts. Lines 459-464
- More detailed suggestions can be made for future studies.
* We have expanded our future research suggestions to include in vivo efficacy studies, pharmacokinetics, and synergistic interactions of the bioactive compounds. Lines 464-467

Reviewer 2 Report
Comments and Suggestions for Authors
In my opinion, the submitted manuscript „ Phytochemical Profiling and Biological Evaluation of Dianthus sylvestris subsp. aristidis: A Chromatographic and Mass Spectrometry Approach
to Uncovering Bioactive Metabolites for Skincare and Diabetes Management” meets aims and scope of „Pharmaceuticals” Journal, section Natural Products, and Special Issue Chromatographic and Mass Spectrometry Techniques to Detect Secondary Metabolites from Natural Products of Pharmacological Interest and may be accepted after the revision.
- The brine shrimp nauplii test is useful for preliminary assessment of cytotoxicity of plant extracts. In further studies, toxicity tests on human cell lines are necessary to evaluate the toxicity of Dinathus sylvestris extracts to humans. In the conclusion of the manuscript, the statement in line 396 is too optimistic, taking into account only the preliminary toxicity tests carried out.
- The Latin terms: in vitro and Dinathus superbus, Dinathus Chinensis, Dinathus carmelitarum, Artemia salina, Helicobacter pylori etc could be written in italics (line 117, 118, 129, 132, 152, 213, 231, 242).
- In Table 1, some names of chemical compounds are written with a lowercase letter, and others with a capital letter - write them consistently in the chosen way.
- Check the grammar of sentences on lines 302, 303, 318.
- "Reference 1" and "reference 2" are missing spaces on line 183.
- In line 38 „aristidis” should be written with a lowercase letter.
Author Response
Reviewer 2 (CHANGES ARE MARKED IN GREEN)
The corrections have been highlighted in the document by a green color
Quality of English Language
(x) The English is fine and does not require any improvement.
|
Yes |
Can be improved |
Must be improved |
Not applicable |
|
|
Does the introduction provide sufficient background and include all relevant references? |
(x) |
( ) |
( ) |
( ) |
|
Is the research design appropriate? |
( ) |
(x) |
( ) |
( ) |
|
Are the methods adequately described? |
( ) |
(x) |
( ) |
( ) |
|
Are the results clearly presented? |
( ) |
(x) |
( ) |
( ) |
|
Are the conclusions supported by the results? |
( ) |
(x) |
( ) |
( ) |
By responding all comments, we have improved all requested sections in table
Comments and Suggestions for Authors
In my opinion, the submitted manuscript: Phytochemical Profiling and Biological Evaluation of Dianthus sylvestris subsp. aristidis: A Chromatographic and Mass Spectrometry Approach
to Uncovering Bioactive Metabolites for Skincare and Diabetes Management” meets aims and scope of „Pharmaceuticals” Journal, section Natural Products, and Special Issue Chromatographic and Mass Spectrometry Techniques to Detect Secondary Metabolites from Natural Products of Pharmacological Interest and may be accepted after the revision.
- The brine shrimp nauplii test is useful for preliminary assessment of cytotoxicity of plant extracts. In further studies, toxicity tests on human cell lines are necessary to evaluate the toxicity of Dinathus sylvestrisextracts to humans. In the conclusion of the manuscript, the statement in line 451 is too optimistic, taking into account only the preliminary toxicity tests carried out.
*We have revised the conclusion to acknowledge the preliminary nature of the brine shrimp assay and highlight the necessity for additional studies to confirm the safety of Dianthus sylvestris subsp. aristidis extracts for human applications. Lines 451-453
- The Latin terms: in vitroand Dinathus superbus, Dinathus Chinensis, Dinathus carmelitarum, Artemia salina, Helicobacter pylori etccould be written in italics (Lines 112,116,117,127,129,133, 148,235, 255, 285)
- *Thank you for your valuable suggestion. We have revised the manuscript accordingly by italicizing all Latin terms (Lines 112,116,117,127,129,133, 148,235, 255, 285)
- In Table 1, some names of chemical compounds are written with a lowercase letter, and others with a capital letter - write them consistently in the chosen way.
* Thank you for your careful review. We have standardized the formatting of all chemical compound names in Table 1 to ensure consistency.
- Check the grammar of sentences on lines 355, 356 and 371
*The grammar of the sentences on lines 355, 356, and 371 has been checked and corrected where necessary
- "Reference 1" and "reference 2" are missing spaces on line 186.
*The missing spaces in 'Reference 1' and 'Reference 2' have been corrected.
- In line 40 “aristidis” should be written with a lowercase letter.
* The correction has been made: "aristidis"

Round 2
Reviewer 1 Report
Comments and Suggestions for Authors
This version can be publish